# Effect of Carbon Nanofibers on the Viscoelastic Response of Epoxy Resins

**DOI:** 10.3390/polym15040821

**Published:** 2023-02-07

**Authors:** Paulo Santos, Abílio P. Silva, Paulo N. B. Reis

**Affiliations:** 1Centre for Mechanical and Aerospace Science and Technologies (C-MAST), University of Beira Interior, 6201-001 Covilhã, Portugal; 2University of Coimbra, CEMMPRE, Department of Mechanical Engineering, 3030-788 Coimbra, Portugal

**Keywords:** epoxy resin, carbon nanofibers, mechanical testing, stress relaxation, creep

## Abstract

Two epoxy resins with different viscosities were enhanced up to 1 wt.%, applying a simple method with carbon nanofibers (CNFs). These were characterized in terms of static bending stress, stress relaxation, and creep tests. In bending, the contents of 0.5 wt.% and 0.75 wt.% of CNFs on Ebalta and Sicomin epoxies, respectively, promote higher relative bending stress (above 11.5% for both) and elastic modulus (13.1% for Sicomin and 16.2% for Ebalta). This highest bending stress and modulus occurs for the lower viscosity resin (Ebalta) due to its interfacial strength and dispersibility of the fillers. Creep behaviour and stress relaxation for three stress levels (20, 50, and 80 MPa) show the benefits obtained with the addition of CNFs, which act as a network that contributes to the immobility of the polymer chains. A long-term experiment of up to 100 h was successfully applied to fit the Kohlrausch–Williams–Watts (KWW) and Findley models to stress relaxation and creep behaviour with very good accuracy.

## 1. Introduction

Epoxy resins, in addition to playing the important role of matrix in composite materials, are an important class of thermosetting polymers due to their excellent mechanical properties, high adhesiveness to many substrates, and good thermal and chemical resistances [1,2,3].

However, due to this cross-linked structure, the epoxy has low toughness, stiffness, and impact resistance [4]. The addition of a small amount of inorganic nanofiller in the epoxy matrix improves the mechanical properties of the net epoxy, particularly its toughness and stiffness [5]. Different studies refer to the benefit of adding micro and nano ceramic particles, as an example, 1 vol.% of Al_2_O_3_ improves by more than 15% the flexural strength and stiffness [6]; 2 wt.% of CaCO_3_ increments the tensile strength and the toughness by 22% and 37%, respectively [7]; the tensile strength increases by 32% and fracture toughness by 45% with the loading of 4 wt.% of TiO_2_ [8]; adding 3 wt.% of SiO_2_, the tensile and impact strength doubles [9,10]; the tensile strength and the fracture toughness increases by 50% and 106%, respectively, with the introduction of 4 wt.% of Fe_2_O_3_ [11].

Moreover, epoxy enhanced by graphene-based nanomaterials has been widely explored and are more attractive because of their unique physical properties [12,13]. It is often functionalized to get the desired characteristics, namely, to improve the surface ionic bonds and the interfacial van der Waals interactions. The carbon nanotubes (CNTs) fundamentally come in three variations: singular-walled CNTs (SWCNTs), double-walled CNTs (DWCNTs), and multi-walled CNTs (MWCNTs). They have a typical one-dimensional (1D) cylindrical fibrous structure, diameters ranging from fractions to tens of nanometres and lengths up to several micrometres and can be considered a cylindrical graphene sheet covered by fullerene-like structures. Their reported surface areas range from 150 to 1500 m^2^ g^−1^, which is a basis for serving as good sorbents [14]. Carbon nanofibers (CNFs) have lengths in the order of micrometres, diameters ranging from ten to hundreds of nanometres (50–200 nm), specific surface area up to 1877 m^2^ g^−1^, and aspect ratio greater than 100. The cylindrical nanostructures of CNFs have a different chemical structure of graphene sheets, such as stacked platelet, ribbon, spiral, fishbone hollow core, fishbone solid, ribbon, stacked cup, and amorphous CNFs without graphene layers [14,15]. Carbon fibres typically used as reinforcements have diameters of several micrometres and are produced from petroleum-based precursors, such as high-strength polyacrylonitrile (PAN) and mesophase pitch (MPP) [16,17].

The relative enhancement in mechanical properties mainly depends on the shape, surface area, and the quality of the interface between matrix and carbon nanofillers. Their large surface-to-volume ratio leads to agglomeration, and their chemically inert nature leads to poor interfacial interactions [18]. Thus, different functionalization and dispersion methodologies correspond to different loadings optimizations and consequently, to different strengths, stiffness, and other enhanced properties. Thus, different wt.% loadings of carbon nanofillers result in different strengths and stiffness levels. For example, reinforcement with CNTs is typically done with loads up to 2 wt.%, but there are also reinforcements carried out with more than 10 wt.% [13]. In many cases, very low CNT loads of 0.1–0.5 wt.% promote very significant improvements: tensile strength and elongation-to-break increases by 12.6% and 25.4%, respectively, with 0.1 wt.% [12]; and tensile strength increases 68% with 0.3 wt.% [13]. In addition, smaller loads facilitate manufacturing, especially in optimizing simple, low-cost, and technologically easy-to-implement techniques.

In previous works, applying a manual lay-up process, known to be a simple method and particularly suitable for large components, it was verified that the use of CNFs benefits the mechanical properties of the epoxy resins [3,19]. In addition, other works with benefits from the use of CNFs in epoxy are referred to in the literature. Shokrieh et al. [20] studied the reinforcing effect of carbon nanofiber (up to 1 wt.% by weight) of an epoxy resin using a mechanical stirrer and sonication. The maximum improvements in tensile strength and flexural strength occurring for 0.25 wt.% of CNFs were 23 and 10%, respectively. Sun et al. [21] carried out a study of the static and dynamic mechanical behaviour of epoxy nanocomposites with CNFs. The results show the highest tensile strength (8%) and the maximum Young’s modulus (17%) are found for 1.0 wt.% of CNFs.

However, most published work characterizes the mechanical performance of these nano-reinforced systems at the level of quasi-static properties and mainly in tensile mode. Nevertheless, in many engineering applications, knowledge of the viscoelastic response (stress relaxation and creep) of materials is crucial due to the requirement for long-term dimensional stability of structures/components. Therefore, in addition to the experimental studies, it is also possible to find several models in literature to predict the viscoelastic response. Although the simplest models are based on a spring or dashpot to represent a purely elastic or ideal fluidic material, they are not applicable to most materials used in engineering because they are neither purely elastic nor ideally fluidic. For these conditions, models involving springs and dashpots with different arrangements are suggested, and among others, it is possible to find the Maxwell model, Voigt model, Boltzmann model, Burgers model, Kelvin model, etc. A review and comparability of these models can be found in [22]. To obtain predictions with better accuracy, and for specific materials, literature also suggests the Kohlrausch–Williams–Watts (KWW) model, an empirical “stretched exponential” function [23].

Therefore, the main goal of this work is to apply a manual lay-up process, known to be a simple, low-cost, and technologically easy-to-implement method, to study the viscoelastic behaviour of two epoxy resins with different viscosities and nano reinforced with low loads of CNFs. Thus, stress relaxation and creep for short periods are assessed through a detailed and comparative methodology of mechanical tests, and the KWW and Findley models are proposed to predict the viscoelastic behaviour of the two nano-enhanced epoxy resins based on experimental static flexural stress results for long-term service.

## 2. Materials and Experimental Procedure

Nanocomposites were produced with two different epoxy resins, which were enhanced with carbon nanofibers (CNFs). For this purpose, an epoxy resin, SR 8100, and a hardener, SD 8822, both supplied by Sicomin, and an epoxy resin, AH 150, and a hardener, IP 430, both supplied by Ebalta, were used due to their different viscosities: Ebalta resin with lower viscosity than Sicomin of 250 ± 50 and 390 mPa × s, respectively, according to the technical data sheet of the resins. In terms of CNFs, they were supplied by Sigma-Aldrich, and according to the manufacturer’s datasheet, they have an average diameter of around 130 nm, a length between 20–200 µm, and an average specific surface area of around 54 m^2^/g. Several contents by weight were studied (0.25, 0.5, 0.75, and 1 wt.%). More details about the resins and CNFs can be found in [3]. For example, for Sicomin resin with 0.75 wt.% CNFs, 150 g of resin, 33 g of hardener, and 1.37 g of CNFs were used. The weight content of CNFs was selected according to the literature and from the perspective of mechanical performance, where typical values of those used in this study can be found [16]. It should be noted the electrical performance is outside the scope of this work because, according to Farzaneh et al. [24], electrical conductivity increases with increasing carbon-based nano reinforcement content due to the formation of a continuous network and the facilitation of free electron mobility.

These nanocomposites were produced by adding CNFs to the epoxy resins (see procedure in Figure 1), and the mixture was carried out at room temperature in a mixer, LBX STIV-020-001, with a shear rate of 1000 rpm for 3 h, followed by 10 min at 150 rpm to disperse the hardener into the system. All of the procedure was combined with sonication, Ultrasonic Cleaner model AU-65, (using an ultrasonic bath with a frequency of 40 kHz) to improve the dispersion of the nanofibers [19,25] in which the temperature was controlled not to exceed the glass transition temperature $\left(T_{g}\right)$ of the resins. Finally, the mixture was degassed in a vacuum oven, Bacoeng Vacuum Chamber, with the aid of a vacuum pump, VEVOR 3CFM, to remove air bubbles and was poured into a cardboard mould with dimensions of 100 × 200 × 3 mm^3^. The nanocomposite-manufacturing process ends with the cure and post-cure suggested in the technical datasheets. While nanocomposites produced with Sicomin resin were cured at room temperature for 24 h and subjected to post-cure at 40 °C for 24 h, those involving Ebalta resin were cured at room temperature for 48 h and subjected to post-cure at 80 °C for 5 h.

The samples used in the experiments were cut from these thin plates into specimens with dimensions of 80 × 10 × 3 mm^3^ (Figure 2a) and tested in a Shimadzu machine, model Autograph AGS-X, equipped with a 10 kN load cell. The static characterization was performed in bending mode and for this purpose, in accordance with the European Standard EN ISO 178:2003; three-point-bending (3PB) static tests were carried out at room temperature and using a span length of 50 mm (Figure 2b). For each condition, at least five specimens were tested at a displacement rate of 2 mm/min, and the main properties for bending strength, bending modulus, and bending strain were obtained according to Equations (1) to (3), respectively.
(1)$$\sigma =\frac{3\;P\;L}{2\;b\;h^{2}}$$
(2)$$E=\frac{\Delta P\;L^{3}}{48\;\Delta u\;I}$$
(3)$$\varepsilon_{f}=\frac{6\;S\;h}{L^{2}}$$
where *P* is the load, *L* is the span length, *b* is the width, *h* is the thickness of the specimen, *I* is the moment of inertia of the cross-section, Δ*P* and Δ*u* are, respectively, the load range and flexural displacement range in the middle span for an interval in the linear region of the load versus displacement plot, and *S* is the deflexion. The error is the standard deviation.

Stress-relaxation tests were performed at room temperature and in accordance with ASTM E328-13 standard in the same machine. A fixed strain was applied (correspondent to 20, 50, and 80 MPa), and the stress was recorded during the loading time (180 min). On the other hand, the creep tests were performed at room temperature and in accordance with ASTM D2990-09 standard where a fixed bending stress was applied (with similar values to those previously reported), and the displacement was recorded during the loading time (180 min). The values used in both tests with at least five specimens were selected to ensure all viscoelastic tests were performed within the elastic regime. The maximum and minimum values reported were as extremes (error) of the solution envelope.

## 3. Results and Discussion

To evaluate the benefits obtained with the CNFs in both resins as well as to select the values used in the creep and stress relaxation tests, static bending tests were carried out. This is important to ensure all tests are performed within the elastic regime of the nanocomposites. Therefore, Figure 3 presents typical bending stress–strain curves for Ebalta resin, which are representative of all curves obtained in this study (including for Sicomin resin).

Regardless of the weight content of CNFs, all curves evidence a linear increase in bending stress with strain, followed by a non-linear behaviour in which the maximum bending stress is reached. For both resins and all conditions analysed, the bending stress decreases until the imminent collapse occurs. It is notorious that the lowest value of the bending strength occurs for neat resin, which increases with the weight content of CNFs until reaching a maximum value. After this maximum, this property is significantly affected by the filler content, and similar behaviour was observed for Sicomin resin but for a different weight content of CNFs.

This evidence can be seen in Figure 4, which summarizes the main bending properties resulting from these curves. Symbols represent the average values and the dispersion bands respective of the maximum and minimum values. Black symbols are related to the bending stress, grey symbols represent the average values of bending stiffness, and blue symbols represent the bending strain. Quantitative analysis for the Sicomin resin (Figure 4a) reveals an increase of about 11.8% in the bending strength when comparing the values obtained for the neat resin (106.2 MPa) and those obtained for 0.75 wt.% (118.7 MPa). Subsequently, the bending strength decreases from 118.7 MPa to 114.2 MPa when the filler content increased up to 1 wt.% of CNFs.

On the other hand, the same analysis for the Ebalta resin shows the maximum bending stress is reached for 0.5 wt.% of CNFs with a value of 123.4 MPa, which is 11.7% higher than that obtained for neat resin (110.5 MPa). Regarding the bending modulus and for both resins, an increase was observed with increasing filler content. While for the Sicomin resin, the increase was around 13.1% and between the neat resin (2.68 GPa) and the nanocomposite reinforced with 1 wt.% of CNFs (3.03 GPa), for the Ebalta resin this value is around 16.2% (from 2.84 GPa to 3.3 GPa). However, when the bending stiffness obtained for the highest bending stress is compared with that of the neat resin, an increase of 11.9% is observed for Sicomin resin and 11.3% for Ebalta. Finally, the bending strain decreases with the increase of filler content for the Ebalta resin, around 6.4% between the value obtained for the neat resin (5.78%) and that of the nanocomposite with 0.5 wt.% of CNFs (5.41%), while for the Sicomin resin, it appears to be constant up to 0.5 wt.% and then decreases.

The reported increase, according to Farzaneh et al. [24], can be explained by the higher modulus of the nanofillers compared to the polymer as well as to the promotion of microphase separation and the formation of harder domains in the presence of the nanofillers. For these authors, the well-controlled microphase separation explains the mechanism responsible for the improvements in strength and modulus. On the other hand, the presence of nanofillers and the formation of the filler network limited the chain mobility and consequently, a decrease in the ultimate bending strain [24]. Furthermore, these results agree with the literature because for higher filler contents, agglomerations/aggregations (corresponding to defects) are observed (see Figure 5), which act as stress concentration points in nanocomposites [26,27,28]. The SEM images shown in Figure 5 refer to the fracture surfaces of the samples tested with the Ebalta resin but are also representative of the Sicomin resin with 0.75 and 1 wt.% of CNFs. A good dispersion of the nano-fillers is evident for 0.5 wt.% of CNFs (Figure 5a), while for 0.75 wt.%, the previously reported agglomerations/aggregations are visible (Figure 5b). Moreover, the interfacial area between the polymer matrix and nanoparticles also decreases and consequently, the mechanical involvement of polymer chains with the nanoparticles [29]. In this context, because only a few polymer molecules can penetrate between the nanoparticles, the viscosity also increases substantially [30]. Finally, according to Fiedler et al. [31], resins with low viscosity promote a better organization of nanoparticles and consequently, better mechanical properties for lower filler contents. This evidence is confirmed in this study because the highest bending stress and modulus were obtained with the Ebalta resin (the one with the lowest viscosity).

Therefore, the proposed study on stress relaxation and creep will only consider configurations that maximized the bending strength. This is due to the fact that higher levels of CNFs promote agglomerations of nanoparticles with a consequent negative effect on the creep response of nanocomposites [32]. In this case, for the creep tests, a fixed bending stress was applied with values of 20, 50, and 80 MPa for both resin and filler contents, and the displacement was recorded during the loading time (180 min). Therefore, from the experimental tests, the curves shown in Figure 6 were obtained where *D* is the bending displacement obtained at any instant of the test, and *D_0_* is the initial bending displacement. These results are representative of the creep behaviour of the other conditions analysed.

All curves clearly show three regions, an instantaneous deformation followed by the first stage and an unfinished secondary one. Inevitably, under these conditions, the third stage is not observed because the creep rupture or failure is out of consideration in this study. In other words, this work focuses on short-term tests that prove to be an easy, fast, and reliable methodology to predict long-term behaviour [33]. The first region is time independent, and the elongation/displacement is attributed to the elastic and plastic deformation of the polymer under a constant applied load but strongly dependent on its magnitude [32,34,35]. For all materials, an increase in instantaneous displacement is evident with increasing applied load as well as for all creep displacements. For example, when comparing the creep displacement between neat resins for the bending stress of 80 MPa, it is observed that after 180 min, the Ebalta resin presents values 14.2% higher than the Sicomin resin. In fact, the creep displacement increases nonlinearly with time, even at room temperature, as shown in Figure 6 and at stress levels much lower than the ultimate strength due to the combination of elastic strain and viscous flow [36,37,38]. However, there is a quantitative relationship between molecular mobility and macroscopic deformation [39], which explains the 14.2% difference reported above. On the other hand, according to Bouafif et al. [40], creep is due to molecular motion in the backbone polymer arrangement and depends on the stress level. This explains the different values observed. For example, comparing the values for 180 min and 20 MPa test, while the creep displacement for the Sicomin resin increases by about 3.5% for 50 MPa and 20.8% for 80 MPa, these values are 6.6% and 31.4%, respectively, for the Ebalta resin. These values also show the higher sensitivity to the loading level of the Ebalta resin in relation to Sicomin one due to its higher molecular mobility [39]. In this context, Vlasveld et al. [41] even reported the deformation process in polymers under load is strongly dependent on the mobility of the chains and not only dependent on temperature.

Another evidence observed in Figure 6 is the benefits achieved by filling the resins with carbon nanofibers. In terms of Ebalta resin, CNFs improve the creep resistance by 10.4%, while for Sicomin resin it is around 1.2%. In this context, the large number of dispersed nanoparticles binds to the matrix via interphase, bridging segments and junctions to support the load and improve the immobility of the polymer chains [35]. This immobility is related to the restriction to slippage, realignment, and motion of polymeric chains that CNFs cause [32], and three mechanisms can contribute to this: (i) good interfacial strength between CNFs/matrix, (ii) CNFs act as blocking sites, and (iii) high aspect ratio of CNFs [42]. Furthermore, below $T_{g}$, the molecular weight has a very small influence on the creep behaviour because only the local motion of the chain segments is involved in the glassy state [41]. Therefore, the higher creep resistance observed for Ebalta resin can be explained by the stronger interfaces that were established between CNFs/polymer. All these facts are summarized in Figure 7 where the difference between initial and final bending displacement (ΔD) is shown. It is possible to compare the load effect on the creep behaviour previously reported, both for neat resins and for the respective nanocomposites as well as the CNFs effect for each resin.

An increase in creep displacement for higher loads is visible as well as increased creep resistance when resins are reinforced with CNFs. However, this difference is more expressive for higher stresses because at lower load levels there are no clearly visible benefits for resins filled with CNFs. According to Yang et al. [35], higher loads increase viscous flow and can even activate non-linear viscoelasticity mechanisms for very high loads. Finally, it is also visible that the benefits achieved for the Ebalta resin are more expressive due to higher interaction between CNFs/polymer.

In terms of stress relaxation, Figure 8 shows the average bending stress versus time curves, where σ_0_ is the bending stress at any given moment of the test, and σ is the initial bending stress. The final values represent the average, maximum, and minimum values obtained for all conditions analysed after 180 min of testing. For comparability, bending displacements corresponding to the same values of bending stress used in the creep tests (20, 50, and 80 MPa) were used. It is notorious that all materials show a decrease in stress over time, and because this study focuses on short-term tests, as reported above, it would not be expected to reach a constant value for bending stress. This will only occur for higher stress values or longer tests.

Another evidence reported by the curves is the existence of an initial stage where the bending stress decreases considerably compared to the remaining time [43,44]. For example, considering only the resins filled with CNFs and bending stress of 80 MPa (Figure 8c,d), the Sicomin one reveals a decrease of about 8.2% in the first 30 min, while the remaining time it decreases (between 30 and 180 min) about 6.3%. These values for the Ebalta resin are 12.5% and 8.5%, respectively. According to the literature, stress relaxation occurs due to physical and/or chemical phenomena. In the first case, it results from molecular rearrangements that require little formation or rupture of the primary bonds, while the second one is due to chain scission, crosslink scission, or crosslink formation [45,46,47]. However, for resins nano-reinforced, all these processes are delayed because the CNFs act as a network that contributes to the immobility of the polymer chains [32,35,42].

Figure 9 compares the differences between initial and final bending stress (Δσ), and it is possible to observe higher stress relaxations for higher bending displacements as well as the benefits achieved with nano-reinforced resins. For example, for Sicomin resin and for the bending displacement corresponding to the highest bending stress (80 MPa), CNFs decreased the stress relaxation by around 21.7% compared to the neat resin, while for the same conditions, this value is about 9.2% for Ebalta resin. It is also noted that with decreasing the bending displacement, the stress relaxation values also decrease, and there is practically no difference between neat and nano-reinforced resins for the smallest bending displacement (corresponding to 20 MPa).

Literature reports several models to predict the viscoelastic response from short-term tests. In terms of creep, Findley’s law is widely used to describe the creep response of composite materials [40,48,49,50] and can even be supported by short-term tests [50,51,52]. The Findley law is given by:(4)$$\varepsilon \left(t\right)=\varepsilon_{0\;}+At^{n}$$
where ε(t) is the creep displacement at time, *t*, $\varepsilon_{0\;}$ is the instantaneous elastic displacement, *A* is the amplitude of transient creep (time-dependent), and *n* is a constant independent of the stress and generally less than one [52]. All parameters were obtained according to the recommendations of Gupta and Lahiri [51]. However, some studies show the KWW model estimates the creep response better than the Findley model [23,53]. In this case, for comparability, this model will also be analysed, which is given by the following equation:(5)$$\varepsilon \left(t\right)=\varepsilon_{0}\;e^{{\left(\frac{t}{\tau }\right)}^{\beta }}$$
where ε(t) is the creep displacement at time, *t*, $\varepsilon_{0}$ is the initial displacement when a constant stress is applied, *β* parameter is the distribution factor related to the breadth of the distribution of creep times, and τ accounts for the mean creep time.

Figure 10 compares the experimental results with those obtained by the two models and evaluates the accuracy of each one in predicting results.

Although the illustrated results refer to Ebalta resin filled with CNFs, they replicate what was observed for the other materials studied. Table 1 and Table 2 present the parameters of each model for all conditions analysed and the accuracy of each model in relation to the experimental results. Therefore, from Figure 10a, it is possible to observe both models fit the experimental data successfully, denoting a maximum error of less than 0.5%. Furthermore, from Table 1 and Table 2, it is possible to conclude the maximum error observed for all conditions studied after 180 min of testing is 2.75%, which evidences the good accuracy obtained. Other evidence taken from these tables is that the most significant errors occur for the highest level of the applied load. Subsequently, to predict the creep response for any bending stress, Figure 10c,d show the parameters of both models versus bending stress for the Ebalta resin with CNFs, and the value of 65 MPa (corresponding to the white marks) to validate the proposed methodology. For the other materials, Table 3 and Table 4 present the respective parameters of each equation. Considering the parameters shown in Figure 10c,d and the respective values presented in Table 3 and Table 4 for 65 MPa, it is possible to obtain the respective values of the equations to predict the creep response. In this case, the following values were obtained: $\varepsilon_{0}$ = 2.93, *β* = 0.395, and τ = 1.60 × 10^4^ for the KWW model and $\varepsilon_{0}$ = 0.760, *A* = 1.716, and *n* = 4.64 × 10^−2^ for the Findley model, and the estimated curves are compared with the experimental ones in Figure 10b).

It is possible to observe both models estimate the bending stress effect on the creep behaviour with good accuracy. After 180 min of testing, the maximum error obtained with the KWW model is 1.5%, overestimating the creep response, and 0.5% for the Findley model. However, for all materials, the maximum error obtained is 4.8% where the Findley model presented the lowest value. Nevertheless, when these models are used to predict long-term creep responses, Figure 11 compares the estimated curves with those obtained experimentally. In this case, 100 h of testing was considered, a value 33 times more than that used in the short-term tests. Although this comparison is made for Ebalta resin with 0.5 wt.% of CNFs and bending stress of 65 MPa, it is representative of the others.

It is possible to observe both models present good accuracy in predicting the creep response after 100 h because the theoretical results obtained are within the dispersion bands resulting from the experimental tests. The errors obtained are around 6.3% obtained with the Findley model and 3% with the KWW model, and while the latter overestimates the experimental result, the Findley model underestimates them. Therefore, the KWW model is more conservative and seems to present better accuracy in relation to the experimental results for longer lives.

In terms of stress relaxation, literature reports several models, but more complex ones than those based on spring-dashpot systems are preferable to obtain predictions with better accuracy [54]. Although the constants have no physical meaning, the Kohlrausch–Williams–Watts model (KWW) can describe the experimental curves very accurately, and it can be used to predict the stress-relaxation response for longer lives [44,55,56]. In this case, the relaxation function *ϕ* is given by:(6)$$\phi \left(t\right)=\frac{\sigma \left(t\right)}{\sigma_{0}}=e^{-{\left(\frac{t}{\tau }\right)}^{\beta }}$$
where σ(t) and $\sigma_{0}$ are, respectively, the stress at time, *t*, and at *t* = 0, *β* is the fractional power exponent, and τ is the KWW relaxation time. In this context, Figure 12 compares the experimental and theoretical curves using the KWW model to evaluate its accuracy.

It is shown the results for Ebalta resin with 0.5 wt.% of CNFs and for the bending displacement corresponding to bending stress of 20 MPa, but they are representative of the other materials studied. Table 5 and Table 6 present the parameters of the model for all conditions analysed and the accuracy of the model in relation to the experimental results. Therefore, from Figure 12a, it is possible to observe a good accuracy where the error between theoretical and experimental curves is only about 0.52% after 180 min of testing. Considering all materials and conditions, from Table 5 it is possible to observe the maximum error is less than 1.3%, which confirms the good accuracy reported above. In addition, to predict the stress relaxation response for any bending displacement, Figure 12c shows the parameters of the model versus bending stress (corresponding to the analysed bending displacement) for the Ebalta resin with CNFs as well as the value of 65 MPa (corresponding to the white marks) to validate the prediction. The values of the equations used to predict the stress-relaxation response are shown in Table 6.

Therefore, the following values were used to compare the theoretical results with the experimental ones: *β* = 0.357, and τ = 2.30 × 10^4^. This comparison shown in Figure 12b reveals a good accuracy where the error for this condition is about 0.32%, but considering all materials, the maximum error observed is less than 3.6%.

Finally, when this model is used to predict the long-term stress-relaxation response, Figure 13 confirms its accuracy by showing the error between the theoretical and experimental values is only 4.9%. Furthermore, the predicted values at the end of 100 h of testing are within the scatter bands, which represent the respective maximum and minimum values obtained from the experimental tests. Therefore, this evidence solidifies the conclusion that the models predict quite well the viscoelastic response of the materials and can be used with good accuracy.

Therefore, the benefits observed with nano-reinforced resins is extremely important because these systems can be transferred to composites where the matrix is the phase with the lowest mechanical performance. This is not only evident for the response to static loads but also for the viscoelastic behaviour of structures/components produced by composite materials. It is conveniently reported in the literature that when the fibres are incorporated into the matrix, they hinder the molecular flow and consequently, delay its viscoelastic response. Regardless of this fact and due to the inherent viscoelasticity of the matrix phase, polymeric composites are prone to creep and stress relaxation, becoming a major challenge when used in long-term applications. However, from this study, it is evident the presence of CNFs delays the viscoelastic response of the resins because they act as a network that contributes to the immobility of the polymeric chains and consequently, decreases the viscoelastic response of the composite materials. Finally, the methodology based on the KWW model presented in this study proves to be effective in predicting the viscoelastic response for long-term applications of structures/components produced by composite materials.

## 4. Conclusions

The main goal of this study is to evaluate the benefits of resins reinforced with carbon nanofibers, and for this purpose, two resins with different viscosities were used.

The static-bending performance showed in both cases, higher values of CNFs added to the resins promoted higher bending stress and modulus; however, an ideal value was observed that maximized these properties. While the optimum weight content was 0.75% for Sicomin SR 8100 resin, for Ebalta AH 150 it was 0.5 wt.%. In addition, the highest bending stress and modulus were obtained with the lower viscosity resin (Ebalta resin) because it promoted a better organization of the nanoparticles. On the other hand, it would be expected the resin with higher viscosity would maximize its mechanical properties for lower filler contents, but the compatibility between the properties of the nanoparticles and matrix significantly influenced the interfacial strength and dispersibility of the fillers.

In terms of creep behaviour and stress relaxation, both phenomena were shown to be strongly dependent on the applied load level. Furthermore, because the study was based on short-term tests, the creep tests presented only the first two regimes, and in the case of the relaxation tests, the stress decrease never reached a constant value for the period under study. However, regardless of the resin, the benefits obtained with the nano-reinforcements were evident because CNFs act as a network that contributes to the immobility of the polymer chains. Finally, for both creep and stress-relaxation behaviour, the results were adjusted following the Kohlrausch–Williams–Watts model, evidencing a good accuracy of the model for longer times in both cases. However, for shorter times, the Findley model shows higher accuracy to estimate the creep behaviour.

## Figures and Tables

**Figure 1 polymers-15-00821-f001:**
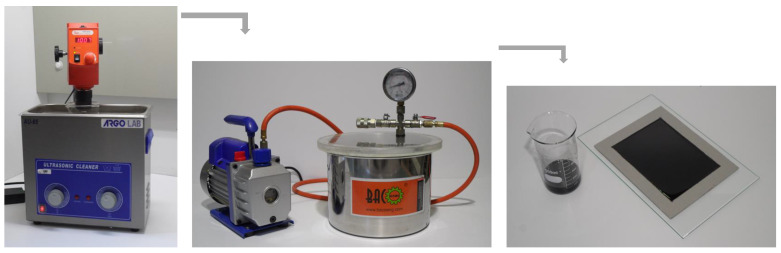
Methodology used to produce the nano-enhanced resins with CNFs.

**Figure 2 polymers-15-00821-f002:**
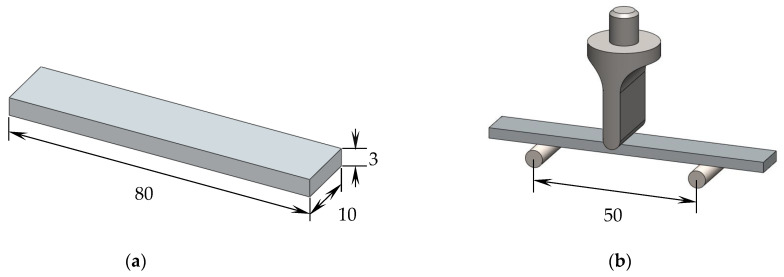
(**a**) Geometry of the specimens; (**b**) schematic view of the 3PB apparatus. All dimensions in mm.

**Figure 3 polymers-15-00821-f003:**
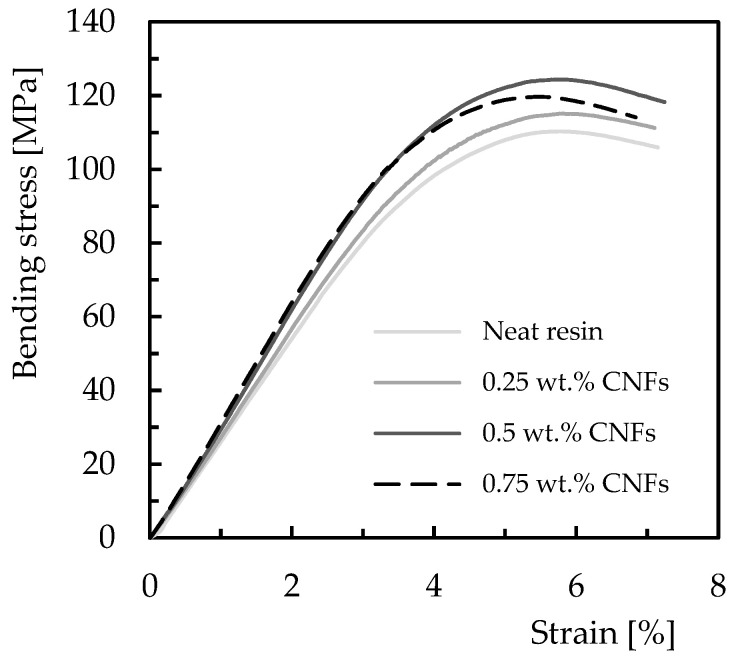
Average flexural stress–strain curves for Ebalta resin with different CNFs contents.

**Figure 4 polymers-15-00821-f004:**
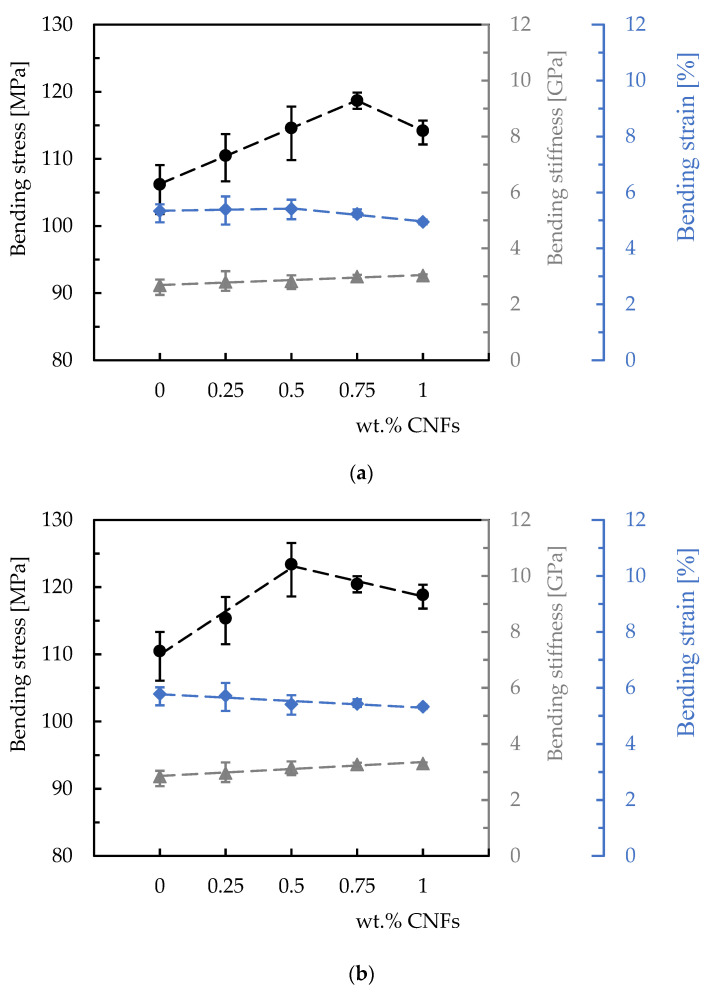
Bending properties versus weight content of CNFs for: (**a**) Sicomin resin; (**b**) Ebalta resin.

**Figure 5 polymers-15-00821-f005:**
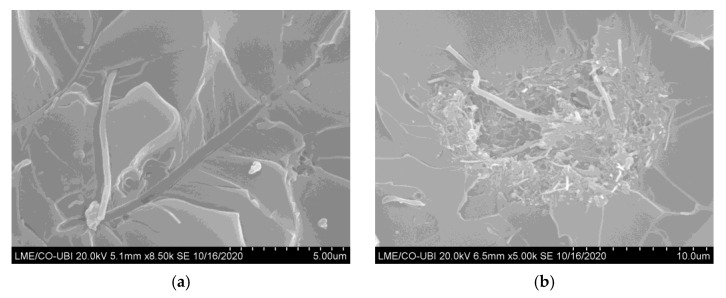
SEM pictures for the Ebalta resin with: (**a**) 0.5 wt.% of CNFs; (**b**) 0.75 wt.% of CNFs.

**Figure 6 polymers-15-00821-f006:**
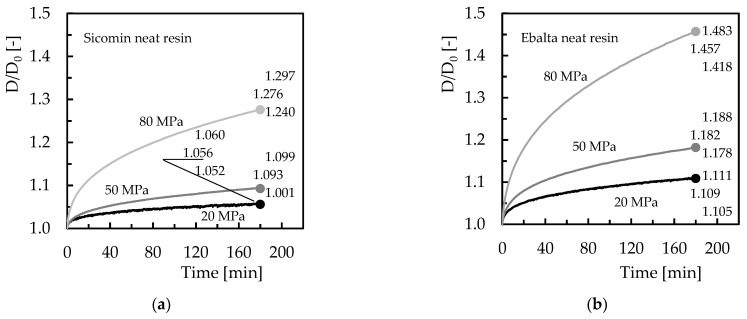
Creep curves for: (**a**) Sicomin neat resin and different bending stresses; (**b**) Ebalta neat resin and different bending stresses; (**c**) Sicomin resin with 0.75 wt.% of CNFs and different bending stresses; (**d**) Ebalta resin with 0.5 wt.% of CNFs and different bending stresses; (**e**) Sicomin neat resin and with 0.75 wt.% of CNFs for bending stress of 80 MPa; (**f**) Ebalta neat resin and with 0.5 wt.% of CNFs for bending stress of 80 MPa.

**Figure 7 polymers-15-00821-f007:**
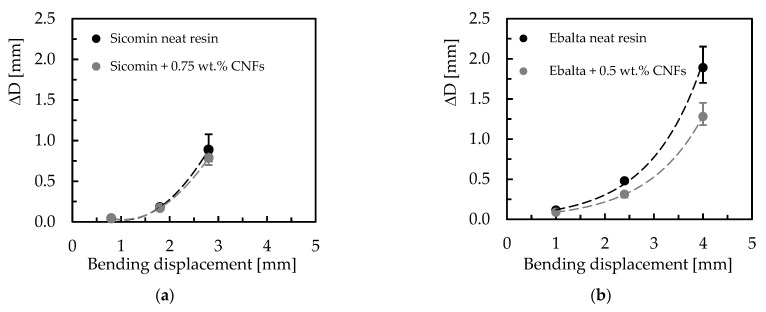
Difference between initial and final bending displacement for: (**a**) Sicomin resin; (**b**) Ebalta resin.

**Figure 8 polymers-15-00821-f008:**
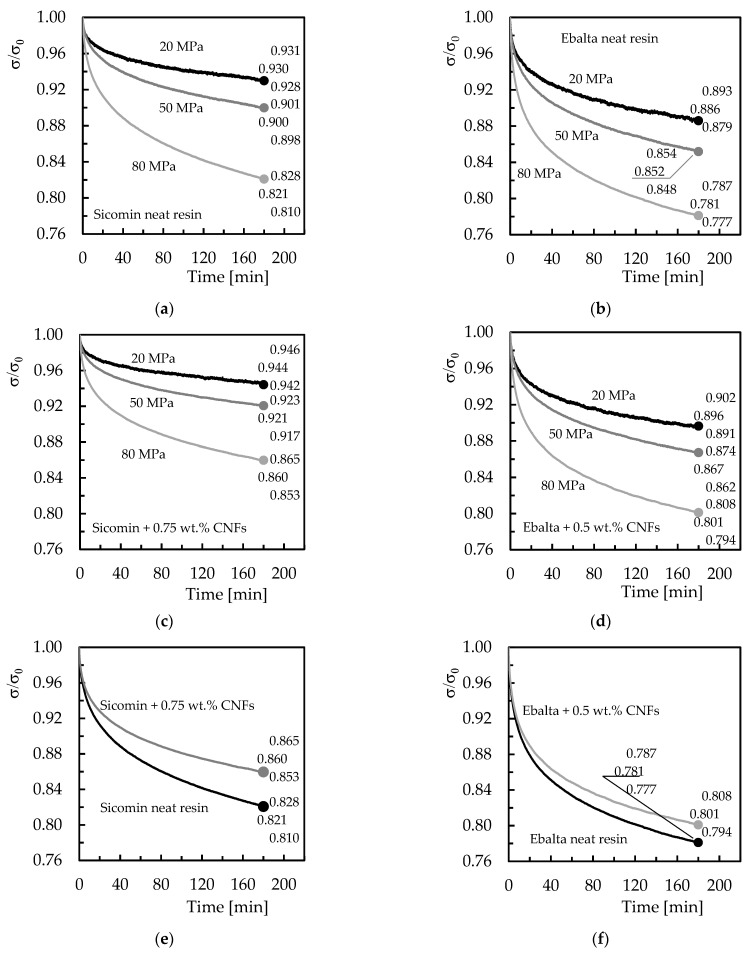
Stress relaxation curves for: (**a**) Sicomin neat resin and different bending stresses; (**b**) Ebalta neat resin and different bending stresses; (**c**) Sicomin resin with 0.75 wt.% of CNFs and different bending stresses; (**d**) Ebalta resin with 0.5 wt.% of CNFs and different bending stresses; (**e**) Sicomin neat resin and with 0.75 wt.% of CNFs for bending stress of 80 MPa; (**f**) Ebalta neat resin and with 0.5 wt.% of CNFs for bending stress of 80 MPa.

**Figure 9 polymers-15-00821-f009:**
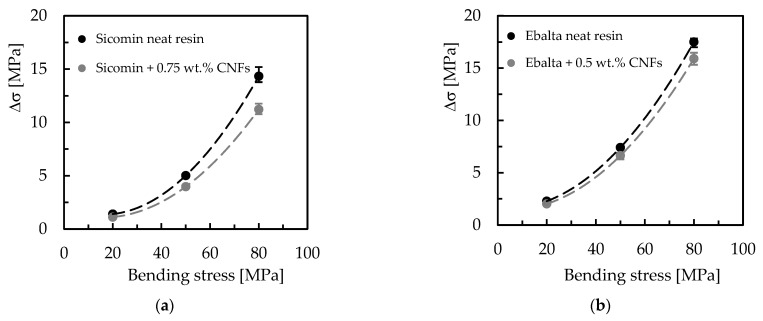
Difference between initial and final bending stress for: (**a**) Sicomin resin; (**b**) Ebalta resin.

**Figure 10 polymers-15-00821-f010:**
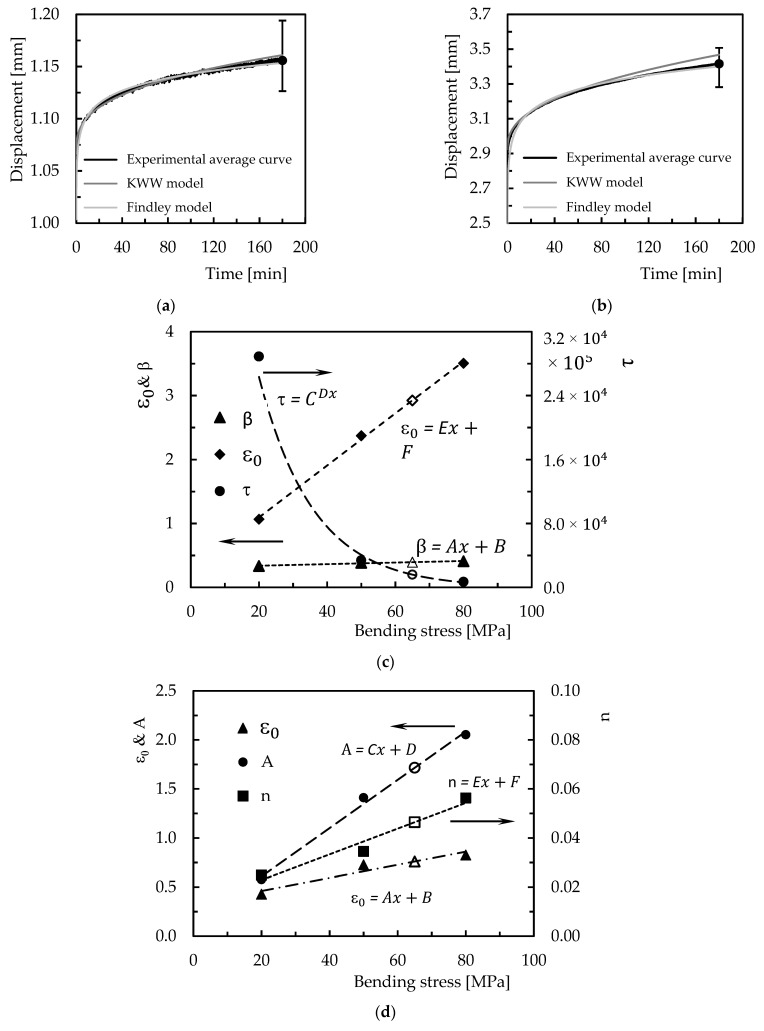
(**a**) Comparison between the experimental and theoretical curves for Ebalta resin with 0.5 wt.% of CNFs and bending stress of 20 MPa; (**b**) model validation for the same material and bending stress of 65 MPa; (**c**) KWW parameters versus bending stress; (**d**) Findley parameters versus bending stress.

**Figure 11 polymers-15-00821-f011:**
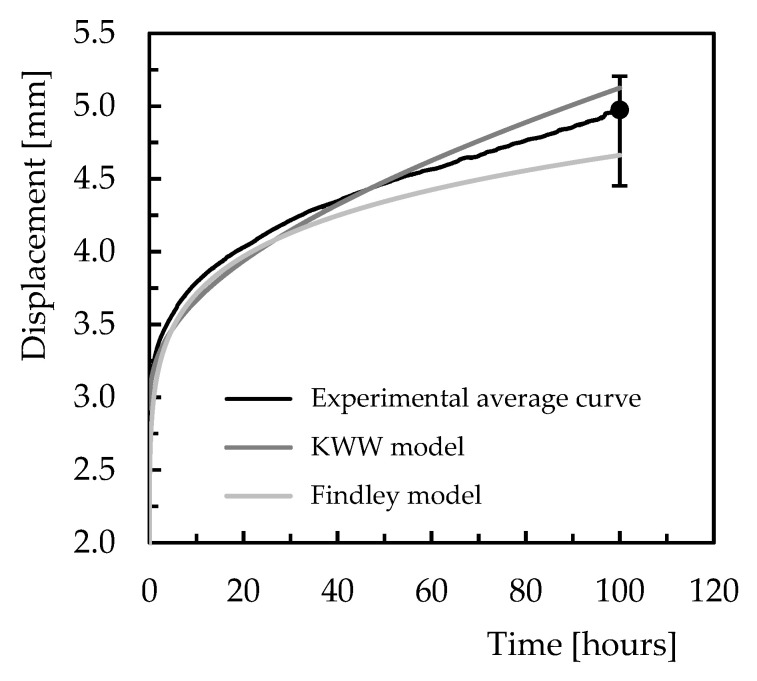
Models’ validation for 100 h (Ebalta resin with 0.5 wt.% of CNFs and bending stress of 65 MPa.

**Figure 12 polymers-15-00821-f012:**
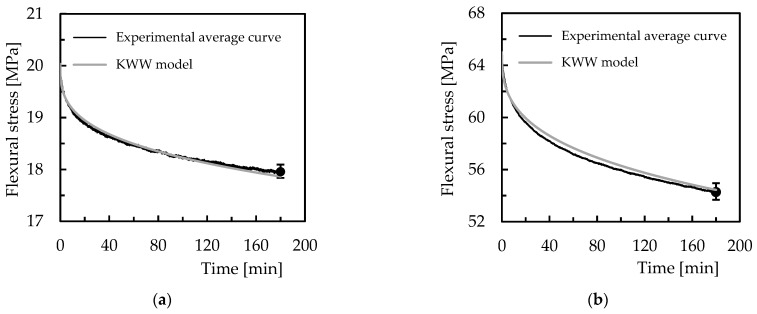
(**a**) Comparison between experimental and theoretical curves for Ebalta resin with 0.5 wt.% of CNFs and bending displacement corresponding to a bending stress of 20 MPa; (**b**) model validation for the same material and bending displacement corresponding to a bending stress of 65 MPa; (**c**) KWW parameters versus bending stress that correspond to the bending displacements studied.

**Figure 13 polymers-15-00821-f013:**
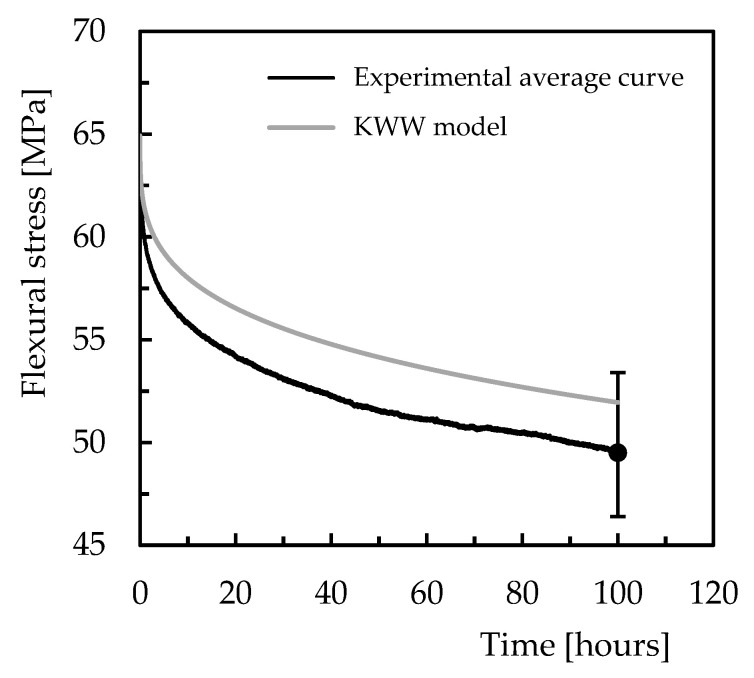
Validation of the model for 100 h and for Ebalta resin with 0.5 wt.% of CNFs and bending displacement corresponding to bending stress of 65 MPa.

**Table 1 polymers-15-00821-t001:** KWW model parameters for creep.

Bending Stress [MPa]	$\varepsilon_{0}$	β	τ	Displacement after 3 h [mm]		
				Experimental Value	KWW Value	Error [%]
Sicomin resin						
20	0.892	0.333	1.37 × 10^6^	0.936	0.939	0.319
50	2.03	0.392	8.94 × 10^4^	2.21	2.22	0.253
80	3.40	0.431	5.12 × 10^3^	4.29	4.31	0.409
Sicomin resin + 0.75 wt.% CNFs						
20	0.765	0.318	1.50 × 10^6^	0.807	0.809	0.216
50	1.81	0.416	5.44 × 10^4^	1.98	1.99	0.437
80	2.86	0.417	4.86 × 10^3^	3.64	3.68	0.949
Ebalta resin						
20	1.06	0.362	8.72 × 10^4^	1.18	1.18	0.36
50	2.67	0.407	1.27 × 10^4^	3.16	3.19	0.954
80	4.02	0.430	1.53 × 10^3^	5.83	5.99	2.75
Ebalta resin + 0.5 wt.% CNFs						
20	1.07	0.335	2.89 × 10^5^	1.16	1.16	0.444
50	2.37	0.385	3.38 × 10^4^	2.68	2.71	0.983
80	3.51	0.409	6.90 × 10^3^	4.34	4.39	1.29

**Table 2 polymers-15-00821-t002:** Findley’s law parameters for creep.

Bending Stress [MPa]	$\varepsilon_{0}$	A	n	Displacement after 3 h [mm]		
				Experimental Value	Findley Value	Error [%]
Sicomin resin						
20	0.406	0.454	0.017	0.936	0.935	0.035
50	0.979	0.873	0.037	2.21	2.21	0.253
80	0.780	1.913	0.063	4.29	4.22	1.67
Sicomin resin + 0.75 wt.% CNFs						
20	0.368	0.367	0.019	0.807	0.806	0.135
50	0.822	0.847	0.033	1.98	1.97	0.600
80	0.792	1.477	0.069	3.64	3.60	1.27
Ebalta resin						
20	0.414	0.563	0.032	1.18	1.17	0.418
50	0.796	1.500	0.048	3.16	3.13	0.829
80	0.932	2.042	0.091	5.83	5.71	2.087
Ebalta resin + 0.5 wt.% CNFs						
20	0.427	0.576	2.49 × 10^−2^	1.16	1.15	0.203
50	0.726	1.410	3.45 × 10^−2^	2.68	2.67	0.536
80	0.827	2.053	5.63 × 10^−2^	4.34	4.29	1.049

**Table 3 polymers-15-00821-t003:** Values of the equations that fit the KWW model.

**Material**	$\varepsilon_{0}$			β			τ		
	*A*	*B*	R	*C*	*D*	R	*E*	*F*	R
Sicomin									
Neat resin	0.042	0.018	0.997	1.64 × 10^−3^	0.304	0.993	9.01 × 10^6^	−0.093	0.999
Resin + 0.75 wt.% CNFs	0.035	0.069	0.999	1.64 × 10^−3^	0.301	0.869	8.74 × 10^6^	−0.096	0.996
Ebalta									
Neat resin	0.049	0.120	0.999	1.13 × 10^−3^	0.343	0.982	3.46 × 10^5^	−0.067	0.999
Resin + 0.5 wt.% CNFs	0.041	0.282	0.999	1.24 × 10^−3^	0.314	0.980	9.15 × 10^5^	−0.062	0.999

R = Correlation coefficient.

**Table 4 polymers-15-00821-t004:** Values of the equations that fit the Findley model.

**Material**	$\varepsilon_{0}$			A			n		
	*A*	*B*	R	*C*	*D*	R	*E*	*F*	R
Sicomin									
Neat resin	6.24 × 10^−3^	0.410	0.643	0.024	−0.136	0.971	7.77 × 10^−4^	3.26 × 10^−5^	0.997
Resin + 0.75 wt.% CNFs	7.06 × 10^−3^	0.308	0.834	0.018	−0.028	0.997	8.35 × 10^−4^	−1.43 × 10^−3^	0.969
Ebalta									
Neat resin	8.64 × 10^−3^	0.282	0.964	0.025	0.136	0.988	9.85 × 10^−4^	7.98 × 10^−3^	0.964
Resin + 0.5 wt.% CNFs	6.66 × 10^−3^	0.327	0.961	0.025	0.116	0.997	5.24 × 10^−4^	1.24 × 10^−2^	0.976

R = Correlation coefficient.

**Table 5 polymers-15-00821-t005:** Parameters of the KWW model for stress relaxation.

Initial Bending Stress [MPa]	β	τ	Bending Stress after 3 h [MPa]		
			Experimental Value	KWW Value	Error [%]
Sicomin neat resin					
20	0.321	6.36 × 10^5^	18.62	18.63	0.012
50	0.388	5.56 × 10^4^	45.07	44.90	0.378
80	0.394	1.15 × 10^4^	66.24	65.88	0.554
Sicomin resin + 0.75 wt.% CNFs					
20	0.312	1.86 × 10^6^	18.94	18.97	0.163
50	0.360	1.69 × 10^5^	46.07	45.96	0.225
80	0.361	3.04 × 10^4^	68.64	68.36	0.059
Ebalta neat resin					
20	0.338	8.57 × 10^4^	17.73	17.68	0.299
50	0.376	2.12 × 10^4^	42.59	42.33	0.622
80	0.349	8.60 × 10^3^	62.47	61.69	1.255
Ebalta resin + 0.5 wt.% CNFs					
20	0.328	1.33 × 10^5^	17.96	17.86	0.520
50	0.361	3.48 × 10^4^	43.32	43.01	0.523
80	0.359	1.66 × 10^4^	66.28	65.66	0.934

**Table 6 polymers-15-00821-t006:** Values of the equations that fit the KWW model.

Material	β			τ		
	*A*	*B*	R	*C*	*D*	R
Sicomin						
Neat resin	1.21 × 10^−3^	0.307	0.897	2.10 × 10^6^	−0.067	0.994
Epoxy + 0.75 wt.% CNFs	8.16 × 10^−4^	0.304	0.874	6.55 × 10^6^	−0.069	0.995
Ebalta						
Neat resin	1.73 × 10^−3^	0.345	0.270	1.19 × 10^7^	−1.64	0.998
Epoxy + 0.5 wt.% CNFs	5.14 × 10^−4^	0.323	0.838	1.19 × 10^7^	−1.49	1.000

R = Correlation coefficient.

## Data Availability

Not applicable.

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
