# Peer review of "Effect of Carbon Nanofibers on the Viscoelastic Response of Epoxy Resins"

_polymers, 2023, doi:10.3390/polym15040821_

Round 1
Reviewer 1 Report
Dear Author,
I studied your manuscript entitled "Effect of carbon nanofibers on the viscoelastic response of epoxy resins". Some spaces need to be improved in terms of journal quality. In my opinion, the technical significance and novelty of the work are still lacking. I recommend major revision before further consideration for publication in the Polymers.
1) The abstract and introduction fail to point out the novelty of the work and its contribution to the state-of-the-art. The authors should clearly state the objectives of their experiments and the motivations driving their research in the first place.
2) How did you select the CNF loadings? Did you measure their percolation threshold? You should spend a few words on this matter.
3) Please present and discuss FTIR analysis for neat and oxidized MWCNTs. XPS curves should also be presented.
4) How did you evaluate the aggregation state of the CNFs? TEM/SEM analyses of samples are very important for verifying the dispersion state of CNFs in the polymer composite. You should provide TEM/SEM images of the samples. I recommend studying the paper " https://doi.org/10.1002/pi.6314" and citing it.
5) English language needs some polishing since some terms are vague.
Author Response
Please, see attached file.

Reviewer 2 Report
1. It is recommended to describe in detail in the Introduction what CNT and CNF are and what their differences are. What is the difference between CNF and conventional carbon fibers?
2. It is recommended in the Introduction to add an overview of the models used to describe viscoelastic behavior.
3. In section 2. Materials and experimental procedure, in line 84, specify where the data for viscosity is taken from.
4. In section 2. Materials and experimental procedure, it is recommended to describe the production of samples in more detail. Which mixer was used? How much material was used? It is recommended to add photos of the sample manufacturing process.
5. In Figure 3, add an axis for Bending strain [%].
6. It is recommended to try using the same vertical axes (the same maximum values) for all the sub-figures in Figure 4. This will better show the difference in resins. You can also try it with Figures 5, 6.
7. It is recommended to add a discussion the significance of the results. How can the obtained results be applied in the engineering practice of designing products using epoxy resins? Is it possible to use the obtained data to perform calculations of complex products using the finite element methods? How are you planning to continue this research?
Author Response
Please, see attached file.

Round 2
Reviewer 1 Report
Dear Authors,
I have recommended the publication of your article as is.